# A Virtual Pressure and Force Sensor for Safety Evaluation in Collaboration Robot Application [note 1]

**DOI:** 10.3390/s19194328

**Published:** 2019-10-07

**Authors:** Heonseop Shin, Sanghoon Kim, Kwang Seo, Sungsoo Rhim

**Affiliations:** Department of Mechanical Engineering, Kyung Hee University, Yongin-si, Gyeonggi-do 17104, Korea; shs0303@khu.ac.kr (H.S.); kimsh83@khu.ac.kr (S.K.); 6333rhkd@khu.ac.kr (K.S.)

**Keywords:** virtual sensor, collision safety, collaborative robot, contact pressure, contact force

## Abstract

Recent developments in robotics have resulted in implementations that have drastically increased collaborative interactions between robots and humans. As robots have the potential to collide intentionally and/or unexpectedly with a human during the collaboration, effective measures to ensure human safety must be devised. In order to estimate the collision safety of a robot, this study proposes a virtual sensor based on an analytical contact model that accurately estimates the peak collision force and pressure as the robot moves along a pre-defined path, even before the occurrence of a collision event, with a short computation time. The estimated physical interaction values that would be caused by the (hypothetical) collision were compared to the collision safety thresholds provided within ISO/TS 15066 to evaluate the safety of the operation. In this virtual collision sensor model, the nonlinear physical characteristics and the effect of the contact surface shape were included to assure the reliability of the prediction. To verify the effectiveness of the virtual sensor model, the force and pressure estimated by the model were compared with various experimental results and the numerical results obtained from a finite element simulation.

## 1. Introduction

As the robotics field experiences exponential growth in use, instances in which robots are installed or operated in the vicinity of human activity, to improve productivity by directly aiding humans with tasks, have increased. Unlike conventional industrial robots, which are installed in closed-off, restricted areas, collaborative robots (also known as cobots), which share workspaces with humans, have much higher potential to contact humans during operation. Considering the velocity, shape, and inertia of the parts that could come into contact with a human, dangerous accidents are likely to occur when robots move too close to a human.

To prevent the serious injury from the human–robot collisions, various studies have been conducted [1,2]. Commercial collaborative robots, in particular, adopt the collision detection technique and deploy a protective stop after a human–robot collision occurs. Generally, these robots use a detection method based on readings from force and torque sensors, as well as a method that uses an algorithm to observe momentum change. Depending on the parameters of the observer algorithm, the control loop time, and the delay of the robot, the detection time is determined. In the case of an impact that has a duration of 10–20 ms, the reaction of the robot, i.e., the protective stop after the contact detection, does not sufficiently contribute to the reduction in contact force because the time from detection to response is too long compared with the time of maximum impact force [3]. In addition to the tardiness of the current protective strategy, it also has another drawback in that it cannot measure or estimate the collision pressure, which has been an increasing focus in recent years as a physical index of the collision safety of collaborative robots [4,5].

The international standard of safety requirement for a collaborative robot, ISO/TS 15066 [6], states the allowable contact force and the collision peak pressure, which should not be exceeded in a human–robot collision. It is obvious that even with the same contact force the peak pressure varies depending on the shape of the impactor (edges, corners, round fillets, etc.). The smaller the contact area, the higher the pressure for the same collision speed and effective mass of the robot. In other words, adding pressure as a new safety index implies a need to consider the contact surface in collision safety assessments.

This paper proposes a virtual sensor approach that computes the expected collision peak force and pressure that would result from a collision between a robot and a human at a given time using an analytical contact model. By predicting the collision pressure and force before the collision, the proposed method guides the robot in managing its task while complying with the ISO/TS 15066 collision safety threshold. This article is an extension of our previously published work [7].

It is possible to evaluate the collision safety using the conventional finite element (FE) simulation approach for particular collision conditions with the given parameters, including the effective mass, the collision velocity, the impactor shape, and the collision direction. A numerical approach for estimating the collision pressure was proposed in [8]. A two-mass system that describes the human and the robot in a collision is modeled with FE simulation software [9,10]. Given, however, that varying the effective mass of the robot as the robot moves, and considering all the shapes of the contact surface, which change depending on the collision direction, leads to difficulty in modeling the FE model, the previously proposed numerical estimation methods are not adequate as a virtual collision sensor for the posture changing robot. Needless to say, the proposed FE methods have a computational cost that is too high to be used in real motion.

In order to use the virtual sensors as a safety sensor, the computation time for estimating the collision pressure and force should be fast. To satisfy this requirement and to reduce the computational costs, a mathematical contact model based on the Hertz contact model has been proposed for use as a virtual sensor. Even though the Hertz contact model and its variations are widely used for calculating contact pressure and force, these models cannot be directly applied to human–robot collisions because of their assumption on the half-space system and the non-linear mechanical properties of human skin [11,12].

In order to overcome these limitations, a one-layered, nonlinear contact model based on penetration was proposed [7,13]. This mathematical model covers the nonlinear elasticity of human skin and also includes several impactor shapes for computation. The proposed method can estimate both the collision pressure and the force at every control time step with a low computational cost. In this paper, we adopt this model as a virtual collision sensor and verify the reliability of the model by comparing the estimation results of the model to the experimental results and the results of a comparable FE model simulation in which both contact nonlinearity and material nonlinearity are included. The effects of various impactor shapes on collision pressure were studied in the experiments detailed in [14,15].

The collision safety of the collaborative robot could be examined by conducting collision tests at its various hazardous configurations. The test method to measure the collision force and pressure for a collaborative robot was proposed in [16]. An experiment was conducted in which a robot collided with a measurement device designed to be dynamically similar to human body parts [17]. Considering the wide spectrum of collision conditions in terms of the varying effective mass of the robot, the wide range of possible collision speeds, and the various shapes of the parts that could collide with the human, the experimental approach could become overwhelming, time-consuming, and expensive. The proposed virtual sensor can also be used as a collision safety evaluation tool for collaborative robots in the place of (or in addition to) the experimental evaluation tests. This paper also presents several simulations for evaluating robot safety.

## 2. Contact Model

### 2.1. One-layered Nonlinear Contact Model: A Virtual Collision Sensor

Commonly, studies calculating the contact force caused by a human–robot collision are based on the Hertz contact model [12]. The widely used Hertzian model operates under the assumptions of elastic strain, continuous surfaces, elastic half-space, and frictionless surfaces. For a more accurate estimation, a penetration-based one-layer, nonlinear contact model has been suggested [7,13]. The characteristics of human skin are highly nonlinear and exhibit greater nonlinearity when considering the depth of the contact area. The proposed nonlinear model improves reliability in its ability to handle nonlinear factors based on penetration-based methods.

Figure 1 shows a deformed bone–skin system that generally represents the forehead or back of the hand where almost no muscle exists. These calculations assumed that bone is a non-deformable body and skin consists of nonlinear, one-dimensional line elements that have an infinite area. Summation of its area results in $\Delta A_{i}$, which is the projected area on an x-y plane. The calculated normal contact force $F_{c}$ with respect to the depth δ can be obtained as
(1)$$F_{c}=\sum_{i=0}^{n}\lambda P\left(\frac{\delta_{i}}{h}\right)\Delta A_{i},\;i=1,2,3...$$
(2)$$P_{c}=\lambda P\left(\frac{\delta }{h}\right)$$

In Equation (1), $\delta_{i}$ is the deformation of the $i^{th}$ line element and P represents the contact pressure of the indentation test with a flat-ended rigid cylinder. Thus, $\lambda P(\delta_{i}/h)$ refers to the pressure corresponding to the strain occurring at the $i^{th}$ line element. $P_{c}$ is the maximum contact pressure corresponding to the depth δ and n is the number of elements from the center to the boundary on the x-z plane. λ is a shape-dependent factor that is obtained by dividing the maximum pressure measured during the indentation test, in which the axis of the cylinder is set parallel to the ground, by that of the flat-ended rigid cylinder, in which the axis of the cylinder is set perpendicular to the ground. The maximum pressure of the cylinder indentation and the maximum pressure of the flat rigid cylinder indentation were introduced in [18,19]. These two values must be found with respect to the same deformation. The thickness of the skin is denoted as h. The model includes surface tension by considering the shape factor λ in Equation (2). However, the change of the contact area due to the surface tension is not taken into account. Sink-down or pile-up phenomena is an example of the surface tension-related effect during the indentation [20]. We assumed that the pile-up and sink-down effects do not significantly affect the contact force because the deformation at the increased or decreased area is relatively less than the maximum deformation.

This method considers only the normal force. The penetration depth in the $i^{th}$ line element is denoted by $\delta_{i}$, which can be obtained by
(3)$$\mathrm{Impactor}\;(\mathrm{cylinder}):\;x^{2}+{\left(z+(R_{r}-\delta )\right)}^{2}={R_{r}}^{2}\;for\;-\infty <y<\infty ,$$
(4)$$\mathrm{Human}\;(\mathrm{sphere}):\;x^{2}+y^{2}+{\left(z-R_{h}\right)}^{2}={R_{h}}^{2}.$$

In order to calculate $\delta_{i}$ from Equations (3) and (4), penetration in the z direction should be derived as
(5)$$\delta_{i}(x,y)=\sqrt{R_{r}-x^{2}}+\sqrt{R_{h}-x^{2}-y^{2}}-R_{r}-R_{h}+\delta ,$$
where $R_{r}$ and $R_{h}$ represent the radius of the collaborative robot and human sample, respectively. In the case of the cylinder and flat contact shown in Figure 1b, $R_{h}=\infty$. Then, Equation (5) becomes
(6)$$\delta_{i}(x)=\sqrt{R_{r}-x^{2}}-R_{r}+\delta .$$

With width w, the contact area can be calculated. Although only the cylindrical impactor type is described in this paper, other types such as sphere and the fillet part of the cylinder-shape impactor will be further studied.

### 2.2. Finite Element Model

To describe the bone and skin system, an FE model was created using the commercial software MIDAS NFX as a one-layered contact model. Focusing on the application where the robot is used in a collaborative operation and moving at a relatively low speed compared to the conventional industrial robots that are used in workspaces completely separated from human workspaces, we investigated collisions that would only cause pain or minor injury, rather than a major bone-related injury to humans. Biomechanical limits introduced in ISO/TS 15066 as a criterion for evaluating the collision safety of robots in this paper are also used as the pain threshold for human body parts, rather than the major injury threshold. For this reason, we assume that the compliance of the bone will not play a significant role in a low-speed collision. Therefore, only the skin was modeled with a hyper-elastic material and the impactor and bone were modeled using rigid body elements. In order to set the FE model and experiment with the same indentation conditions, the bone was fixed to the ground. The impactor shape was identified as a cylindrical body, as shown in Figure 2. For efficient analysis, the non-deformable part of the impactor was modeled using rigid shell elements, and the system was displayed with a 1/4 symmetric scheme. The diameter of the impactor was set to 20 mm and the thickness of the silicone rubber was set to 4 mm. In this simulation, the width of the silicone rubber was 40 mm. As shown in Figure 2, the FE model was designed with a rigid cylinder and a flat rubber surface ($R_{h}=\infty$) on the shell. In order to describe the reliability of the large deformation analysis, the center parts of the rubber were modeled with a relatively fine mesh. Additionally, hybrid elements and a compatible implicit integration scheme were used [21].

## 3. Validation of Mathematical Models Comparing with FE Model and Experiment

For validation of the mathematical model and the FE model, hyper-elastic silicone rubber indentation experiments were conducted. In this experiment, flat rubber was used for the validation of the cylinder-flat contact case shown in Figure 1b. The size of the rubber used in the experiment was 50 mm × 40 mm with a thickness of 4 mm. Figure 3a illustrates the indentation experimental setup. To measure the peak contact pressure by Tekscan, a film-type mapping pressure sensor was attached to the impactor as shown in Figure 3a. Figure 3b,c shows the actual silicone rubber deformation caused by an impactor compared to the FE model. The experiment was conducted in the same manner as the conventional indentation test. To minimize the effect of friction between the rubber, the impactor, the base plate, and the surface were lubricated with silicone oil.

In the mathematical model, which was written in MATLAB, n was set to 30 and multiplier λ was set to pi/2 [20]. The same values for $R_{r}$, $R_{h}$, h, and w were used in the experiments and the FE model. The compression direction was always normal to the rubber surface during the compression test. Figure 4 shows the indentation force and the peak pressure of the FE simulation and the mathematical model with respect to normalized displacement compared to those obtained in the indentation experiment. The contact force and pressure obtained using the proposed mathematical model and FE model show good agreement with those obtained in the experiments.

The calculation time of the mathematical model written in MATLAB is 0.13 s for 100 sampling points during compression until normalized displacement reaches approximately 0.72. This is notably quicker than the calculation time of the FE model of 5520 s for 170 sampling points during compression until normalized displacement reaches approximately 0.62 using an Intel i7-7700, 16GB RAM computer system. In addition, the force and pressure can be easily calculated with the mathematical model in MATLAB for various collision events by changing the parameters of the model such as the thickness of the skin, the geometry (shape) of the impactor, and the collision direction. These benefits make the mathematical model attractive for applying to a robot as a virtual force/pressure estimation sensor. This technique is useful for operating the robot in a safe manner by controlling the speed of the robot in advance or by warning the human if the monitored value is higher than a set threshold.

## 4. Evaluation of Robot Safety

### 4.1. Dynamic Modeling of Human–Robot Collisions

The peak collision pressure and force is affected by several parameters, such as the effective masses of the robot and the human, the collision direction, the impactor geometry, the thickness of the human skin and the impact velocity. As these variables continuously vary depending on the human–robot collision event, it is necessary to calculate the contact force and pressure at a low computational cost for the safety evaluation in real-time. A human–robot collision dynamic model consists of a two-mass system given by the equation below:(7)$$\left[\begin{array}{cc} M_{r}(\gamma ) & 0\\ 0 & M_{h} \end{array}\right]\left\{\begin{array}{c} {\ddot{y}}_{r}(t)\\ {\ddot{y}}_{h}(t) \end{array}\right\}=\left\{\begin{array}{c} -F_{c}(t)\\ F_{c}(t) \end{array}\right\}$$
(8)$$\delta (t)=y_{r}(t)-y_{h}(t)$$
(9)$$F(\gamma )=\max(F_{c}(t))$$
(10)$$P(\gamma )=\max(P_{c}(t))$$
(11)$$0\leq t\leq t_{end}$$

Here, γ is the normalized position of the end-effector of the robot along the given path (0≤γ≤1). t implies the simulation time of a human–robot collision that is hypothetically assumed to occur at the position γ. The effective mass of the robot at γ was represented by $M_{r}(\gamma )$ and was calculated based on the inertia matrix of the robot and collision direction [22]. The applied effective mass of the human body part in this simulation, $M_{h}$ represents the effective mass of the human body part. In the case of constrained transient contact, $M_{h}$ becomes infinity. In the following simulation, as we evaluate the robot collision safety assuming that the robot has the possibility of colliding with the forehead in an unconstrained transient contact, $M_{h}$ is set to 4.4 kg [6]. Respectively, $y_{r}$ and $y_{h}$ denote the displacement of the robot and the human. The peak contact force F(γ) and pressure P(γ) at γ are calculated by Equations (1) and (7)–(10). The initial condition of the system is ${\dot{y}}_{r}(0)=v_{r}(\gamma )$, where $v_{r}(\gamma )$ is the normal direction velocity of the impactor to the contact surface at γ. Since vertical contact causes a higher contact force and pressure of the skin than tangential contact, only the normal direction velocity is considered as the collision velocity in this paper. In this simulation, the damping effects were not taken into account for the simplicity of the model and the analysis. When the transferred energy from the robot to the human during the collision is studied in the future, the damping effects will be considered in the extended human–robot collision model.

In the simulation, the time step size of integration for solving Equation (7) was set to 0.2 ms, and $t_{end}$, the time at which the simulation terminates, was set to 20 ms, as the robot–human head collision lasts only 4–10 ms [3]. MATLAB obtained the peak force F(γ) and pressure P(γ) in 0.59 s. The nominal response time of a typical safety-rated commercial laser sensor satisfying the performance level of ISO 13849-1 is 0.08 to 0.68 s, depending on the sampling rate chosen [23]. Given that the calculation time of the virtual sensor for predicting the peak physical value is lower than the maximum response time of the typical commercially available safety sensor, the proposed virtual sensor (even in the MATLAB environment) can be adapted to the robot system for the evaluation of the collision safety. Considering the limited computational performance of MATLAB, which is relatively slow, we could also drastically decrease the calculation time in the C-language environment.

### 4.2. Evaluation of the Robot Collision Safety

Using the dynamic Equations (7), the peak collision force F(γ) and pressure P(γ) that would result from the collision if it took place at position γ on a path were calculated. The allowable pressure and force threshold were set by the biomechanical limits given in ISO/TS 15066. In the standard, the allowable force and pressure limits for 29 body parts are listed in terms of the collision types (quasi-static and transient collision). By comparing these limits with the estimated values, the collision safety of the robot can be evaluated while traveling along the given (or any arbitrary) trajectory, as shown in Figure 5 and Figure 6. In the simulation, the model of Universal Robot UR5 was used.

Figure 5 shows the three (initial, intermediate and terminal) postures of the robot moving along the given trajectory. The end-effector moves along the straight line between the initial posture and the terminal posture. The three circle-labeled points of the robot, A and B on the third link and C on the fifth link, are the points at which the collision with the human was assumed during the motion. As it is shown in Figure 5, both of the links have a cylindrical shape with a radius of 37.5 mm. The collision part on the human was chosen as the forehead. Thanks to previous research [24], the material properties of the soft tissue of a pig were used in this simulation as a replacement material property of the soft tissue of the forehead of the human. Considering that the geometry of the forehead is very similar to a sphere, the cylinder-to-sphere contact model, as described in Equation (5), was used in this simulation. $R_{h}$ was set to 91.4 mm and h was set to 5.3 mm in Equations (1), (2) and (5) [25,26].

Figure 6 shows the results of the safety evaluation simulation when the robot moves in the given trajectory described in Figure 5. The inertial resistance to the external force of point A is primarily determined by the first and second joint position, and the movement of point A is limited (or constrained) in a particular direction at a particular posture of the robot. For this, the effective mass (which is a function of the posture and the point of the interest [22]) of point A drastically increases at γ=0.42, as shown in Figure 6a. As illustrated in Figure 6c,d, however, the collision force and pressure (which are the function of effective mass and the collision speed) of point A would remain lower during motion because of the slow speed.

Due to the high velocity at point C, the collision force and pressure at point C are higher than the collision force and pressure at other points over the entire trajectory. Among the collisions to the forehead that would occur at the three points during the motion, some of the collisions that would occur at point B and point C, as shown in Figure 6c, will result in contact forces or contact pressures that slightly or significantly exceed the threshold value for the forehead, listed in ISO/TS 15066. In ISO/TS 15066, the transient threshold for the forehead is not specified because collisions with the head are not allowed because of the high risk. Therefore, the quasi-static value of the biomechanical limit for the forehead was chosen as the threshold. Although the quasi-static threshold is much lower than the transient threshold, the contact force due to the collision at point C is 3.38 times the threshold. From these results, we can anticipate that the collision at point C might cause injury to a human. Therefore, it is necessary to control the speed of the robot and/or alter the posture of the robot (to reduce the effective mass) to keep the potential collision force lower than the limit.

## 5. Conclusions

The proposed virtual sensor computes the expected peak collision force and pressure of a collaborative robot colliding with a human. The model assumes that the robot link makes impact with the human during motion. This model takes into consideration the fact that the mechanical characteristics of human skin are highly nonlinear, and its contact area varies depending on the shape of the impactor, the direction of the impact, and the deformation behavior of the body. The force and pressure values estimated by the proposed method were compared to the results of the cylinder indentation experiments and the simulation results from a well-established FE model. It was shown that the estimated values agree well with the experimental data.

The feasibility of the use of the contact model as a virtual collision sensor was illustrated in a simulation where a collaborative robot moves along a pre-defined path, and the estimated peak force and pressure that would result from a hypothetical collision was calculated during the motion. By comparing the estimated values with allowable values stipulated by ISO/TS 15066, the collision safety of the robot was evaluated. The simulation results also particularly note that even with the highly nonlinear nature of the mathematical model, the required time for the calculation of the proposed model is significantly lower (0.004%) than that of the FE model. The computation time to obtain the estimated peak force and pressure, assuming the collision is 0.59 s in the MATLAB environment, and the evaluation of collision safety can be performed as frequently as it is needed in real-time as the robot moves along the given trajectory. Additionally, the collision of a robot with, not only the forehead but also other parts of the human body, can be easily simulated by using the advantage of the mathematical model.

The virtual collision sensor proposed in this paper can be used in conjunction with the control system not only to prevent the human from approaching the robot but also to prevent the robot from moving to collide with the human without reducing its speed. Additionally, it can also be utilized in conjunction with different control or maneuver strategies to enhance the collaborative performance of the robot while ensuring safety. With a controller with the proposed virtual sensor integrated, safe motion planning will be implemented in a real robot system in future works. Even though only the results from the simulation of the cylindrical impactor with a particular radius is presented in this paper, other shapes (e.g., cylinder with smaller or larger radius, wedges, rounded corners) can be dealt with in the model in theory. The future study will look into the shape dependency in collision safety for robots and end-effectors with various shapes.

## Figures and Tables

**Figure 1 sensors-19-04328-f001:**
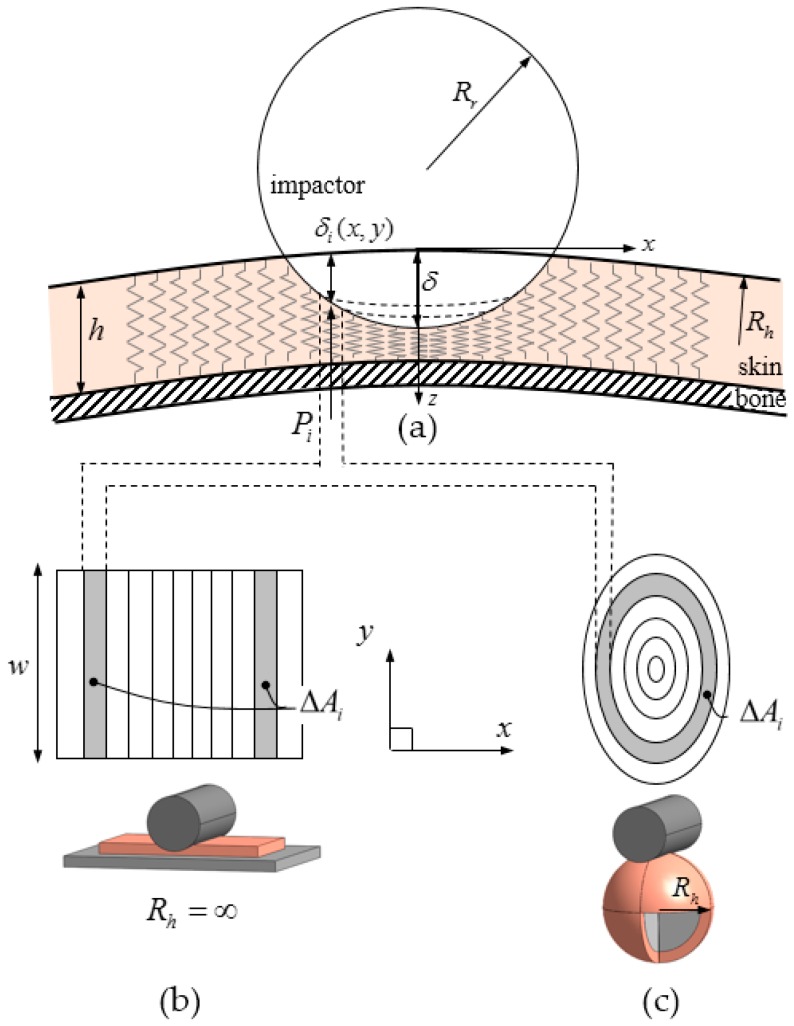
Schematic description of the proposed one-layered contact model: impactor and human-skin contact model (**a**), cylinder-flat contact (**b**), cylinder-sphere contact (**c**).

**Figure 2 sensors-19-04328-f002:**
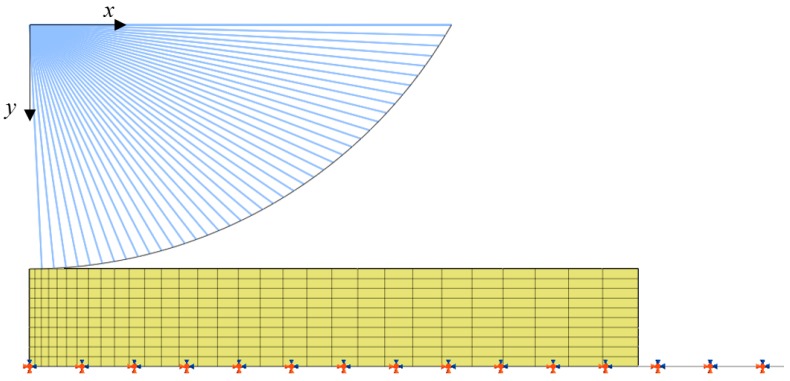
FE model of cylindrical impactor type.

**Figure 3 sensors-19-04328-f003:**
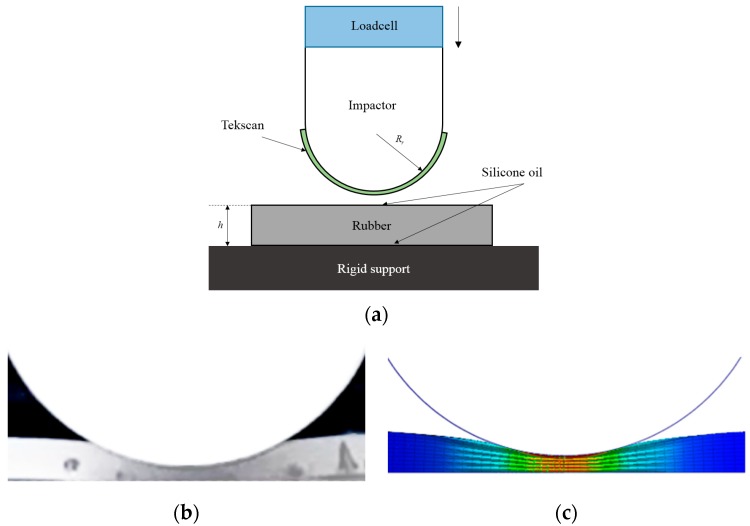
Indentation of silicone rubber: experimental test configuration (**a**), experiment (**b**), FE simulation (**c**).

**Figure 4 sensors-19-04328-f004:**
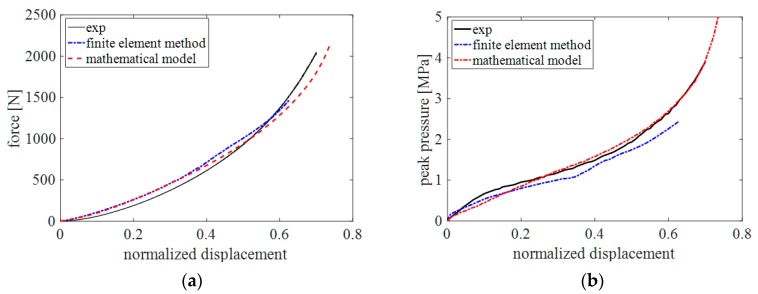
Comparison of mathematical model, FE model and experiment: contact force (**a**) and peak pressure (**b**).

**Figure 5 sensors-19-04328-f005:**
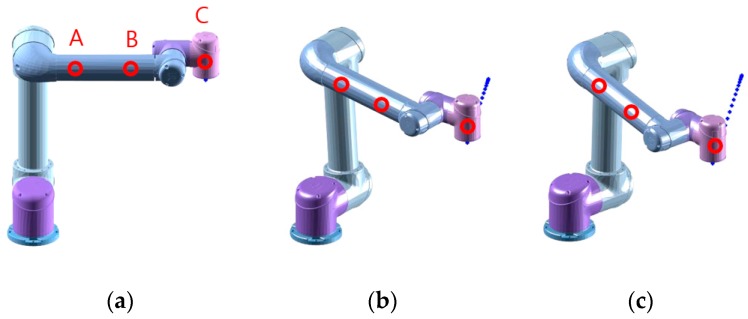
Simulation results: trajectory of the robot: γ=0 (**a**), γ=0.5 (**b**), γ=1 (**c**).

**Figure 6 sensors-19-04328-f006:**
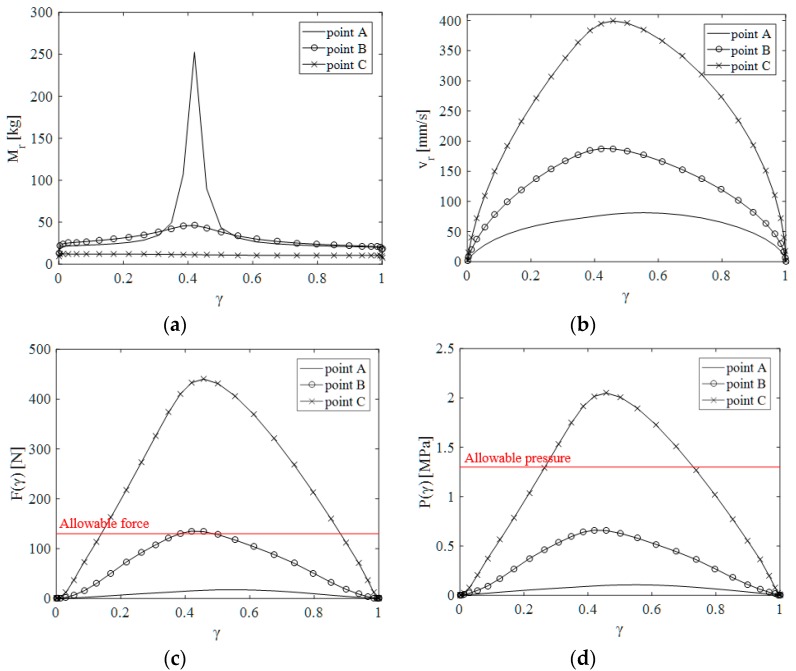
Simulation results for evaluation collision safety of UR5 robot: (**a**) effective mass, (**b**) normal direction velocity to contact surface, (**c**) estimated collision force, (**d**) estimated collision pressure.

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
