# Peer review of "A Virtual Pressure and Force Sensor for Safety Evaluation in Collaboration Robot Application†"

_sensors, 2019, doi:10.3390/s19194328_

Round 1
Reviewer 1 Report
Review of SENSORS 574378 (2019)
Comments to authors:
This paper presents the algorithms and computational load to calculate the danger of a robot moving towards a nearby human in a collaborative human-robot task environment.
General: If implemented as a warning or E-stop-producing system for a robot, assuming that the position of the human is known in relation to the robot, then this could be a valuable method to quantify the danger and provide context-aware warnings or reaction.
It would strengthen the paper to indicate how such a warning system could be incorporated in a robot controller.
Specific comments:
Line 133: The UR5 can achieve speeds of 180°/s. At a radius of 1 m, that’s ~3 m/s. This cannot be considered “low speed”.
Line 134: There is no support provided for the statement, “the model assumes that the stiffness of the bone will not play a notable role during the collision.” The disparity of speeds is mentioned, but a robot moving at 1 m/s, and a human moving at the same speed in the opposite direction can still induce quite an impact load. Please provide support for your simplification.
Line 139: The head is chosen as the candidate body part, but the silicone skin model is 4 mm thick on top of the bone. Please provide biomechanics evidence or support that this is realistic, since the head has been chosen as the impact zone.
Line 141: This reviewer is not convinced that modelling the human as an infinite-radius (flat) surface is valid. It is likely that a human head or arm is the contact point with a robot, and human and robot will both have approximately the same effective radius. I appreciate that the cylinder-on-sphere model is also being considered, but the cylinder-on-plane simplification should perhaps not even be considered as realistic. Please provide support for this assumption.
Line 152: I think you mean “silicone oil”.
Line 168: The result of an impending collision will be an E-stop command to the robot or a warning to the human. If the calculation of impact danger is 0.13 sec, and if a robot is moving at a slow 10 cm/sec, and since human reaction time is ~1 sec at best (hearing a warning and physical reaction to it), then the robot will still move ~11 cm. If the computation time is 0.59 sec (line 197) then the robot will move a ~16 cm. Is this realistic as a safety system? Please justify your conclusion. Will the calculation improvement suggested in Line 198 be sufficient?
Line 220: This paragraph has fractured grammar and is not understandable. Please reword.
Fig. 2. It would be helpful to have axes indicated.
All figures: Please place figures in the document after they are first cited in the text.
English: Before publication, this paper needs a complete rewrite by a native English technical writer. There are many mistakes, leading to confusion on meaning.
Author Response
Dear reviewer,
We are grateful for your consideration of this manuscript, and we also very much appreciate your suggestions, which have been very helpful in improving the manuscript.
We also thank the reviewers for their careful reading of our text.
All the comments we received on this study have been taken into account in improving the quality of the article, and we present our reply to each of them separately. With regard to some of the suggestions, we would note the following:
----------------------------------------------------------------------------------------
[Note]
Figure 6 is changed due to a mistake of calculation v0 of point C. Some sentences are also modified in according to the change made in Figure 6.
It would strengthen the paper to indicate how such a warning system could be incorporated in a robot controller.
[Answer]
Thanks for your suggestion. We would add a sentence about its application to a warning system.
[Add] [Line 293 of modified file]
The virtual collision sensor proposed in this paper can be used in conjunction with the control system not only to prevent the human from approaching the robot but also to prevent the robot from moving to collide with the human without reducing its speed.
Line 133: The UR5 can achieve speeds of 180°/s. At a radius of 1 m, that’s ~3 m/s. This cannot be considered “low speed”.
Line 134: There is no support provided for the statement, “the model assumes that the stiffness of the bone will not play a notable role during the collision.” The disparity of speeds is mentioned, but a robot moving at 1 m/s, and a human moving at the same speed in the opposite direction can still induce quite an impact load. Please provide support for your simplification
[Line 133 of original word file]
Considering a collaborative robot moving at a relatively low speed compared to the speed of conventional industrial robots, the model assumes that the stiffness of the bone will not play a notable role during the collision.
[Answer]
As you mentioned, even the collaborative robot is capable of moving at very high speed. However, in this paper, we focus on the relatively weak collision that would only cause pain or minor injury in the human skin. In order to clarify the scope of the current study we modified the paragraph as follows.
[Modification] [Line 143 of modified file]
Focusing on the application where the robot is used in the collaborative operation and moving at a relatively low speed compared to the conventional industrial robots which is used in the workspace completely separated from the human workspace, we look into the collision that would only cause pain or minor injury rather than the major bone-related injury to human. Biomechanical limits introduced in ISO/TS 15066 as a criterion for evaluating the collision safety of robot in this paper are also given as the pain threshold for human body parts not as the major injury threshold. For this reason we assume that the compliance of the bone will not play a significant role at the low speed collision.
Line 139: The head is chosen as the candidate body part, but the silicone skin model is 4 mm thick on top of the bone. Please provide biomechanics evidence or support that this is realistic, since the head has been chosen as the impact zone.
[Line 139 of original word file]
The diameter of the impactor was set to 20 mm and the thickness of silicone rubber was set to 4 mm.
[Answer]
The thickness of the forehead in collision safety simulation (Figure 5) is chosen according to the reference [26]. It is also stated in Line 248. It would support why the thickness of the silicone rubber was set to 4mm.
Line 141: This reviewer is not convinced that modelling the human as an infinite-radius (flat) surface is valid. It is likely that a human head or arm is the contact point with a robot, and human and robot will both have approximately the same effective radius. I appreciate that the cylinder-on-sphere model is also being considered, but the cylinder-on-plane simplification should perhaps not even be considered as realistic. Please provide support for this assumption.
[Line 141 of original word file]
As shown in Figure 2, the FE model is designed with a rigid cylinder and a flat rubber surface on the shell.
[Answer]
The cylinder-to-flat model is only used for the validation of the proposed model by comparing the force and peak pressure of FE model and experiment. In the collision safety simulation, the cylinder-to-sphere model is used for representing the robot-to- forehead collision. To clarify the condition, we have added a sentence as follows.
[Modification] [Line 249 of modified file]
Considering that the geometry of the forehead is very similar to the sphere, the cylinder-to-sphere contact model as described in equation (5) is used in this simulation.
Line 152: I think you mean “silicone oil”.
[Line 152 of original word file]
silicon oil.
[Answer]
Yes. Thanks for informing us of it. We corrected it.
[Modification]
silicone oil.
Line 168: The result of an impending collision will be an E-stop command to the robot or a warning to the human. If the calculation of impact danger is 0.13 sec, and if a robot is moving at a slow 10 cm/sec, and since human reaction time is ~1 sec at best (hearing a warning and physical reaction to it), then the robot will still move ~11 cm. If the computation time is 0.59 sec (line 197) then the robot will move a ~16 cm. Is this realistic as a safety system? Please justify your conclusion. Will the calculation improvement suggested in Line 198 be sufficient?
[Line 197 of original word file]
MATLAB obtained the peak force and pressure in 0.59 seconds.
[Answer]
To elaborate the fact that the computation time of the virtual sensor is short enough to be adopted as a safety sensor in a real-robot system, we compared the response time of the proposed virtual sensor with that of a typical safety laser scanner sensor commercially available in the robot system.
[Add] [Line 224 of modified file]
In case of a typical safety-rated commercial laser sensor satisfying the performance level of ISO 13849-1, the nominal response time is 0.08 to 0.68 seconds depending on the sampling rate chosen [23]. Given that the calculation time of the virtual sensor for predicting the peak physical value is lower than the maximum response time of the typical commercially available safety sensor, the proposed virtual sensor (even in the MATLAB environment) can be adopted to the robot system for the evaluation of the collision safety. Considering the limited computational performance of MATLAB, which is relatively slow, we could also drastically decrease the calculation time in C-language environment.
Line 220: This paragraph has fractured grammar and is not understandable. Please reword.
[Line 220 of original word file]
Figure 6 shows the results of the safety evaluation simulation. As it is located near the third joint the dynamic behavior of point A is primarily determined by the first and the second joints leads relatively weak redundancy at point A throughout the whole trajectory compared to the speeds of the other two points. As a result, the effective mass (which is a function of the posture and the point of the interests [18]) of point A dramatically increases, as shown in Figure 6a, as it moves to the non-redundant posture.
[Answer]
Thanks for informing us. We corrected it as follows.
[Modification] [Line 253 of modified file]
The inertial resistance to the external force of point A is primarily determined by the first and second joint position and the movement of point A is limited (or constrained) in a particular direction at a particular posture of the robot. For this, the effective mass (which is a function of the posture and the point of the interests [22]) of point A drastically increases at $\gamma = 0.42$, as shown in Figure 6a.
Fig. 2. It would be helpful to have axes indicated.
[Answer]
Thanks for your suggestion. Axes are added.
All figures: Please place figures in the document after they are first cited in the text.
[Answer]
Thanks for your suggestion. We modified the position of figures.
Reviewer 2 Report
General
very interesting and timely work designed to enable online "safety awareness" of robots in collaborative applications, an essential functionality for ease of deployment of well-protected applications some minor typographical errors and minor grammatical imperfections, but these do not impair understanding of contentIntroduction
good overview good motivation for present work Question of energy dissipation in transient contact might be good to mention, though Hertzian contact does not accommodate dissipation Experiments in [15] do NOT include collisions with human subjects. Only collisions between robots and a measurement apparatus were conducted. The apparatus was designed to approximate the reactive inertia of a human body region.Contact model
The descriptions are complete though slightly terse. It took me some time to verify the formulae, but all the information is there. Perhaps a slight bit more detail on the origin of the factor $\lambda$ would be helpful to the reader. The model of the elastic contact shown in Fig. 1 does not include any surface tension of the skin, i.e. the skin touches the impacting cylinder at every point where the cylinder displaces skin from the initial situation. In Fig. 3(b) however, the FE calculations show the more realistic result in which the sides are "pulled down" by surface tension, an effect which can tend to reduce the instantaneous contact area. A comment would be good to take a position on whether or not this effect is negligible and why. Given the interest in the community on the topic of energy transfer, energy deposition in the skin / soft tissue, on power and on power flux density, the authors are invited to add some comments to indicate the scope of this work and whether or not it can be extended to include considerations of energy transfer or perhaps considerations of conservation of linear momentum. It is recognized that this can be quite complex when a driven device such as a robot possibly continues to introduce energy into the system during the contact of interest.Validation of mathematical model
The results in Fig. 4 do look very attractive and should give the authorsconfidence regarding their approach. On the other hand, as is often neglected in the engineering world, some sample error bars or a statement on the respective uncertainties in the results for each of the methods would be an appropriate completion of the presentation of these results. Since many researchers have used the Tekscan sensing technology, they might be thankful for some words on the reproducibility and reliability of the system in this particular experimental arrangement. A photograph of the experimental arrangement used to obtain the measured force and pressure values would be appropriate. Of technical interest to other researchers might be the method or sensor or technology used to measure the (normalized) displacement in the experiments. If there are data available with different radii of impacting cylinder, with different thickness of skin equivalent material / silicone rubber, perhaps even material samples with different properties, this would further shore up the results presented.Evaluation of robot safety
While the model set forth in eqs. (7-11) includes the general case of motion of the human body region, this is not treated in the further discussion. Tacitly it seems to be assumed to be negligible or irrelevant. Please add a description of the choices made here. Further up in the paper it is mentioned that the bone is mechanically grounded. This is interpreted to mean that it is fixed in space and that the contact situation considered is of the type "constrained transient contact". On the other hand, the reactive inertia of the head is mentioned, so the case might indeed be "unconstrained transient contact". The assumptions made should be stated more clearly and the usual terminology should be chosen to support the reader. Please clarify a bit more what exactly the case is. The results presented in Figs. 5 and 6 are fairly clear and straightforward to understand. It takes a moment to understand that the TCP seems to running a linear motion in Fig.5 (a-c). Perhaps additional description can help for easier readability. The very high value of the robot effective mass at contact point A for $\gamma = 0.5$ is surprising in Fig. 6(a), especially since the nearby point B does not have anywhere near as high a value for $M_r$. Please double check $M_r$ at A for $\gamma = 0.5$. The thresholds indicated by the red lines in Fig. 6 (c) and (d) are the force and pressure limits of ISO/TS 15066 for the forehead. Since the values in the TS are for the quasi-static contact, perhaps a sentence or two is appropriate that these values have been taken to gauge the transient case as well in this work, even if the TS does not allow in practice any contact whatsoever with this body area. It is OK to use this example in the paper, but it would help to state the limits of applicability.Conclusion
The authors certainly understand that their methodology is very likely applicable also to other body regions. It would be a good idea for them to address this with comments on possible challenges and limitations of such an extension of the work.Author Response
Dear reviewer,
We are grateful for your consideration of this manuscript, and we also very much appreciate your suggestions, which have been very helpful in improving the manuscript.
We also thank the reviewers for their careful reading of our text.
All the comments we received on this study have been taken into account in improving the quality of the article, and we present our reply to each of them separately. With regard to some of the suggestions, we would note the following:
----------------------------------------------------------------------------------------
[Note]
Figure 6 is changed due to a mistake of calculation v0 of point C. Some sentences are also modified in according to the change made in Figure 6.
Question of energy dissipation in transient contact might be good to mention, though Hertzian contact does not accommodate dissipation
Given the interest in the community on the topic of energy transfer, energy deposition in the skin / soft tissue, on power and on power flux density, the authors are invited to add some comments to indicate the scope of this work and whether or not it can be extended to include considerations of energy transfer or perhaps considerations of conservation of linear momentum. It is recognized that this can be quite complex when a driven device such as a robot possibly continues to introduce energy into the system during the contact of interest.
[Answer]
Thanks for your suggestion. As you mentioned, energy transfer from the robot to the human during a human-robot collision is quite interesting index to look into. We have been studying the index but have not reached a concrete conclusion yet and we will look into that aspect further and the results would be reported in the future publication. We add a sentence about this plan.
[Add] [Line 217 of modified file]
In this simulation, the damping effects are not taken into account for the simplicity of the model and the analysis. When the transferred energy from robots to the human during the collision would be studied in the future , the damping effects will be considered in the extended human-robot collision model.
Experiments in [15] do NOT include collisions with human subjects. Only collisions between robots and a measurement apparatus were conducted. The apparatus was designed to approximate the reactive inertia of a human body region.
[Line 85 of original word file]
The experiments to find out the biomechanical limits of human to the impact were performed by Matthias, in which the human subject was collided with a flat cylinder [15].
[Answer]
Thanks for informing us. We changed the sentence as follows.
[Modification] [Line 87 of modified file]
An experiment was conducted in which a robot collided with a measurement device designed to be dynamically similar to human body parts [17].
The descriptions are complete though slightly terse. It took me some time to verify the formulae, but all the information is there. Perhaps a slight bit more detail on the origin of the factor $\lambda$ would be helpful to the reader.
[Line 114 of original word file]
$\lambda$ is a multiplier obtained by dividing the maximum pressure measured during the indentation tests of a cylinder by that of a flat-ended rigid cylinder.
[Answer]
Thanks for your suggestion. We changed the description of it from a multiplier to a shape factor. Additionally we add a reference [20] for more detail.
[Modification] [Line 117 of modified file]
$\lambda$ is a shape-dependent factor which is obtained by dividing the maximum pressure measured during the indentation test, in which the axis of the cylinder is set parallel to the ground by that of the flat-ended rigid cylinder, in which the axis of the cylinder is set perpendicular to the ground. The maximum pressure of cylinder indentation and the maximum pressure of flat rigid cylinder indentation are introduced in [19, 20].
The model of the elastic contact shown in Fig. 1 does not include any surface tension of the skin, i.e. the skin touches the impacting cylinder at every point where the cylinder displaces skin from the initial situation. In Fig. 3(b) however, the FE calculations show the more realistic result in which the sides are "pulled down" by surface tension, an effect which can tend to reduce the instantaneous contact area. A comment would be good to take a position on whether or not this effect is negligible and why.
[Answer]
Thanks for your comment. This model takes into account the surface tension by multiplying the shape factor $\lambda$ in equation (2). However, the change of contact area due to the surface tension is not taken into account. Sink-down or pile-up phenomena is the example of the surface tension related effect during the indentation. We assumed that the pile-up and sink-down effects dose not significantly contribute on the contact force because the deformation at the increased or decreased area is relatively lower than the maximum deformation. For better understanding, we add a few sentences about it.
[Add] [Line 122 of modified file]
The model includes the surface tension by considering the shape factor $\lambda$ in equation (2). However, the change of the contact area due to the surface tension is not taken into account. Sink-down or pile-up phenomena is the example of the surface tension related effect during the indentation [18]. We assumed that the pile-up and sink-down effects dose not significantly affect the contact force because the deformation at the increased or decreased area is relatively lower than the maximum deformation.
The results in Fig. 4 do look very attractive and should give the authors confidence regarding their approach. On the other hand, as is often neglected in the engineering world, some sample error bars or a statement on the respective uncertainties in the results for each of the methods would be an appropriate completion of the presentation of these results. Since many researchers have used the Tekscan sensing technology, they might be thankful for some words on the reproducibility and reliability of the system in this particular experimental arrangement. A photograph of the experimental arrangement used to obtain the measured force and pressure values would be appropriate. Of technical interest to other researchers might be the method or sensor or technology used to measure the (normalized) displacement in the experiments.
[Answer]
Thanks for your suggestion. We added a figure and sentences that describe the experiment setup
[Add] [Line 165 of modified file]
Figure 3(a) illustrates the indentation experimental setup. To measure the peak contact pressure by Tekscan, a film-type mapping pressure sensor, was attached to the impactor as shown in Figure 3(a).
If there are data available with different radii of impacting cylinder, with different thickness of skin equivalent material / silicone rubber, perhaps even material samples with different properties, this would further shore up the results presented.
[Answer]
Thanks for your comment. We are planning to perform other experiments with various shapes in the future.
Further up in the paper it is mentioned that the bone is mechanically grounded. This is interpreted to mean that it is fixed in space and that the contact situation considered is of the type "constrained transient contact". On the other hand, the reactive inertia of the head is mentioned, so the case might indeed be "unconstrained transient contact". The assumptions made should be stated more clearly and the usual terminology should be chosen to support the reader. Please clarify a bit more what exactly the case is.
[Line 188 of original word file]
The applied effective mass of the human body part in this simulation $\M_h$ is 4.4 kg which is representing the head [4]
[Answer]
Thanks for informing us. We added sentences describing that the collision safety evaluation in section 4. was conducted assuming that the robot collides with the forehead in an unconstrained transient contact situation.
[Modification] [Line 208 of modified file]
The applied effective mass of the human body part in this simulation. $\M_h$ represents the effective mass of the human body part. In the case of constrained transient contact, $\M_h$ becomes infinity. In the following simulation, as we evaluate the robot collision safety assuming that the robot has possibility to collide with the forehead in unconstrained transient contact, $\M_h$ is set to 4.4 kg [6].
The results presented in Figs. 5 and 6 are fairly clear and straightforward to understand. It takes a moment to understand that the TCP seems to running a linear motion in Fig.5 (a-c). Perhaps additional description can help for easier readability.
[Answer]
Thanks for your suggestion. A sentence is added about it.
[Add] [Line 243 of modified file]
The end-effector moves along the straight line between the initial posture and the terminal posture.
The very high value of the robot effective mass at contact point A for $\gamma = 0.5$ is surprising in Fig. 6(a), especially since the nearby point B does not have anywhere near as high a value for $M_r$. Please double check $M_r$ at A for $\gamma = 0.5$.
[Answer]
Thanks for your comments. We have checked it again. The effective mass of point A is drastically increased because the determinant of Jacobian matrix becomes zero at that posture. For better understanding, we modified sentences as follows.
[Modification] [Line 253 of modified file]
The inertial resistance to the external force of point A is primarily determined by the first and second joint position and the movement of point A is limited (or constrained) in a particular direction at a particular posture of the robot. For this, the effective mass (which is a function of the posture and the point of the interests [22]) of point A drastically increases at $\gamma = 0.42$, as shown in Figure 6a.
The thresholds indicated by the red lines in Fig. 6 (c) and (d) are the force and pressure limits of ISO/TS 15066 for the forehead. Since the values in the TS are for the quasi-static contact, perhaps a sentence or two is appropriate that these values have been taken to gauge the transient case as well in this work, even if the TS does not allow in practice any contact whatsoever with this body area. It is OK to use this example in the paper, but it would help to state the limits of applicability.
[Answer]
Thanks for your suggestion. We added sentences describing it.
[Add] [Line 265 of modified file]
In ISO/TS 15066, the transient threshold for the forehead is not specified because the collision with the head is not allowed for the high risk, so the quasi-static value of the biomechanical limit for the forehead was chosen as the threshold. Although the quasi-static threshold is much lower than the transient threshold, the contact force due to the collision at point C is 3.38 times the threshold. From those results, we can anticipate that the collision at point C might cause injury to a human.
The authors certainly understand that their methodology is very likely applicable also to other body regions. It would be a good idea for them to address this with comments on possible challenges and limitations of such an extension of the work.
[Answer]
Thanks for your suggestion. We added sentence describing it.
[Add] [Line 290 of modified file]
Additionally, the collision of a robot with not only the forehead but also other parts of the human body can be easily simulated by using the advantage of the mathematical model.
Reviewer 3 Report
This paper mainly focused on the virtual sensor based on an analytical contact model, which estimates accurately the peak collision force and pressure as the robot moves along a pre-defined path even before the occurrence of a collision event with the short computation time. This paper has enough novelty to be published in this journal. Simulation results discussed suitably. However, I have some concerns about the different parts of the manuscript.
In the introduction section, the main contribution should be concluded point by point. Also briefly summarize the structure of the article;
What is the FE model? Please give a specific name before using the abbreviation;
How do the results in Figure 4 be obtained? Please give details;
Give more details about the simulation analysis and the relationship of the data about Figure 5 and Figure 6;
Please give specific future work according to your research results; Please check carefully the word and grammar errors of the paper;
It is suggested to add more related references in the manuscript, for example: "Fuzzy approximation-based adaptive backstepping control of an exoskeleton for human upper limbs." IEEE Transactions on Fuzzy Systems 23.3 (2014): 555-566; "Safety-enhanced collaborative framework for teleoperated minimally invasive surgery using a 7-DoF torque-controlled robot." International Journal of Control, Automation and Systems 16.6 (2018): 2915-2923. "Improved Human–Robot Collaborative Control of Redundant Robot for Teleoperated Minimally Invasive Surgery." IEEE Robotics and Automation Letters 4.2 (2019): 1447-1453. etc..
Author Response
Dear reviewer,
We are grateful for your consideration of this manuscript, and we also very much appreciate your suggestions, which have been very helpful in improving the manuscript.
We also thank the reviewers for their careful reading of our text.
All the comments we received on this study have been taken into account in improving the quality of the article, and we present our reply to each of them separately. With regard to some of the suggestions, we would note the following:
----------------------------------------------------------------------------------------
[Note]
Figure 6 is changed due to a mistake of calculation v0 of point C. Some sentences are also modified in according to the change made in Figure 6.
What is the FE model? Please give a specific name before using the abbreviation;
[Answer]
Thanks for your comments. We modified to introduce the full term before using the abbreviation
[Modification] [Line 59 of modified file]
It is possible to evaluate the collision safety using the conventional finite element (FE) simulation approach for a particular collision conditions with the given parameters including the effective mass, the collision velocity, the impactor shape, and the collision direction.
How do the results in Figure 4 be obtained? Please give details;
[Answer]
Thanks for your comments. We modified the sentence to describe in detail.
[Modification] [Line 174 of modified file]
Figure 4 shows the indentation force and the peak pressure of the FE simulation and the mathematical model with respect to normalized displacement compared to those obtained in an indentation experiment.
Give more details about the simulation analysis and the relationship of the data about Figure 5 and Figure 6;
[Answer]
Thanks for your comments. We added a sentence to describe in detail.
[Add] [Line 252 of modified file]
Figure 6 shows the results of the safety evaluation simulation when the robot moves in the given trajectory described in Figure 5
Please give specific future work according to your research results
[Answer]
Thanks for your comments. We added a sentence to describe in detail.
[Add] [Line 297 of modified file]
With a controller with the proposed virtual sensor integrated, the safe motion planning will be implemented in real robot system in future works.
It is suggested to add more related references in the manuscript
[Answer]
Thanks for your comments. We added the two references.
[Add] [Line 35 of modified file]
To prevent serious injury from human-robot collision, various researches were conducted [1-2].